# Hip Joint Stability during and after Femoral Lengthening in Congenital Femoral Deficiency

**DOI:** 10.3390/children11040500

**Published:** 2024-04-22

**Authors:** Jędrzej Tschurl, Milud Shadi, Tomasz Kotwicki

**Affiliations:** Department of Spine Disorders and Pediatric Orthopedics, Poznań University of Medical Sciences, 61-701 Poznań, Poland

**Keywords:** hip instability, femoral lengthening, congenital femoral deficiency

## Abstract

Background: Hip stability remains a major preoccupation during femoral lengthening in Congenital Femoral Deficiency (CFD). We aimed to review hip stability in Paley type 1a CFD patients undergoing femoral lengthening. Methods: A total of 33 patients with unilateral CFD, who were treated between 2014 and 2023, were retrospectively reviewed. In 20/33 cases (60.6%) the SUPERhip preparatory surgery was performed at a mean age of 4.3 years (range 2.7–8.1). The femoral lengthening using an external fixator was performed at a mean age of 7.8 years (range 4.3–14.3). Results: All patients presented with a stable hip joint after preparatory surgery and during femoral lengthening. Six cases of hip instability at a mean of 637 days after the external fixator removal were observed (range 127 to 1447 days). No significant differences between stable and unstable hips were noted for (1) Center-Edge Angle: 23.7 vs. 26.1 deg; (2) Acetabular Inclination: 12.8 vs. 11.7 deg; and (3) Ex-Fix Index: 35.6 days/cm vs. 42.4 days/cm; *p* > 0.05. Late hip instability was related to Coxa Vara and decreased femoral antetorsion before lengthening. Conclusions: Late hip joint instability in Paley type 1a CFD patients may occur long after femoral lengthening despite hip morphology appearing to be normal on radiograms before and at the end of femoral lengthening. Coxa Vara, femoral torsional deformity, and posterior acetabular deficiency might be risk factors for hip instability.

## 1. Introduction

Congenital Femoral Deficiency (CFD) is a rare (incidence rate of 1–2 cases per 100,000 births) congenital disorder characterized by a shorter, abnormally formed femur which implies pathological hip and knee joints [1]. The clinical presentation is heterogeneous—it may be an isolated defect or accompany other deformities, unilateral or bilateral, with varying severity from mild shortening affecting the proximal part of the femur to severe, including absence of the femur bone. In its unilateral form, due to limb length discrepancy, CFD is treated with limb lengthening [2,3,4].

CFD affects other structures in addition to the femur. The presence of deformities depends on the severity of CFD, and may include the acetabulum, muscles of the lower limb, blood vessels and the ligamentous apparatus of the knee joint [2,3]. Abduction and flexion contracture of the hip, proximal femur deformities, and torsional deformities (diminished antetorsion or retrotorsion of the femur) may occur in CFD. In some cases, the proximal femur can present delayed ossification or pseudoarthrosis at the neck or subtrochanteric levels. The severity of CFD determines the treatment plan for the patient [5,6]. Currently, the most used classification for CFD is one proposed by Paley in 1998 [2,7]. It is based on the evaluation of the development and ossification of the proximal femur. Each type has a recommended scheme for treatment. For type 1a (normal ossification of the proximal femur), the SUPERhip procedure (Systematic Utilitarian Procedure for Extremity Reconstruction) is proposed by Paley [2,5]. SUPERhip consists of lengthening the soft tissues to correct flexion and abduction contracture; proximal femoral osteotomy (PFO) to correct varus, flexion, and torsional femoral deformity; and pelvic osteotomy whenever needed to correct acetabular dysplasia. SUPERhip is a preparatory surgery to achieve a stable hip joint for the next stages in CFD treatment—femoral lengthening to achieve equal limb length.

CFD treatment remains challenging and linked with additional surgical procedures. Hip instability is a substantial complication that may occur during femoral lengthening in CFD. Hip joint subluxation/dislocation occurring during femoral lengthening was described in the literature [8,9,10,11,12,13,14,15,16]. Some of the authors have proposed risk factors of the developing subluxation or dislocation during femoral lengthening such as severity of CFD, residual acetabular dysplasia, Coxa Vara deformity, proximal femoral osteotomy for the distraction osteogenesis, and Paley type 1b (delayed ossification of the proximal femur). Addressing those risk factors is crucial to lowering the incidence of hip joint instability in CFD.

Reconstruction of acetabular dysplasia in CFD needs to take into consideration studies regarding acetabular morphology, such as research by Dora et al. or Musielak et al. who have demonstrated that the acetabulum in CFD is different from those of patients with developmental dysplasia of the hip and smaller, posteriorly reversed, and more inclined than those in healthy hips [17,18]. Currently used methods of acetabular reconstruction, that try to address those issues, are based on Dega or modified Dega osteotomies [2,19,20].

The purpose of this study is to review a cohort of homogenous Paley type 1a CFD patients to analyze the hip joint stability during and after femoral lengthening. This study aims to analyze the possible causes of hip joint instability and its management.

## 2. Materials and Methods

### 2.1. Material

Approval from the institutional bioethics committee was obtained to carry out this study. We have performed a retrospective review of 50 patients (53 femurs) presenting with CFD and treated between 2014 and 2023. The diagnosis of CFD was made by a pediatric orthopedic surgeon with experience in congenital limb deformities based on clinical and radiological appearance. The inclusion criteria were as follows: (1) type 1 CFD according to Paley classification, (2) availability of full radiographic and clinical data, (3) no previous surgical treatment, (4) two-stage reconstruction surgery consisting of hip reconstruction followed by femoral lengthening using an external fixator, and (5) minimum 1-year follow-up after external fixator removal. Based on these criteria, we have excluded the following: 9 patients (12 femurs) who have not undergone femoral lengthening yet; 3 patients who have previously had surgical treatment (hip joint reconstruction or femoral lengthening) performed in another department (they were referred to our department to treat complications); and 5 patients with Paley type 2 CFD (3 with 2a, 1 with 2b and 1 with 2c type). The flowchart presenting the inclusion process is presented in Figure 1. Finally, the analysis concerned 33 patients (37 femoral lengthenings), all with CFD type 1a according to Paley classification.

All patients presented unilateral CFD while 24/33 patients presented concomitant fibular hemimelia in the form of moderate fibular hypoplasia (18/24; 75%) [21]. A total of 20 patients (22 femoral lengthenings) underwent the SUPERhip procedure as a hip preparatory surgery while the remaining 13 patients (15 femoral lengthenings) were not treated with the SUPERhip procedure before femoral lengthening. The mean follow-up after femoral lengthening was 3 years and 3 months.

Acetabular reconstruction with Dega iliac osteotomy [19] was performed as a part of the SUPERhip procedure, whenever the hip joint radiological morphology showed signs of instability risk, as proposed by Eidelman et al., Suzuki et al., Salai et al., and Bowen et al. [8,9,10,11]: Acetabular Inclination (AI) > 25 deg, Center-Edge Angle (CEA) < 20 deg. Varus deformity of the proximal femur (Neck Shaft Angle, NSA < 120 deg) or femoral antetorsion decreased below 20 degrees were corrected at the time of hip reconstruction. Proximal femoral osteotomies were fixed with a blade plate which was removed one year after hip reconstruction.

From 22/33 (67%) patients who underwent preparatory surgery before lengthening, 20 patients underwent full SUPERhip procedure at mean age of 4.3 years old and subsequent femoral lengthening at mean age of 6.2 years old while the 2 remaining patients underwent hip preparatory surgery limited to proximal femoral valgus and rotation osteotomy at mean age of 4.2 years old, followed by femoral lengthening at mean age of 7.5 years old. The remaining 11/33 (33%) patients were not qualified for hip preparatory surgery based on the stability criteria [8,9,10,11]. These patients had their first femoral lengthening at a mean age of 10 years old. All 22 patients had proximal femoral osteotomy at the time of hip preparatory surgery to address varus and torsional deformity. In all, 16/22 patients (71.4%) had pelvic osteotomy—all of them had Dega osteotomy at the time of hip preparatory surgery.

Femoral lengthening was performed at least one year after plate removal, with the use of an external fixator (monolateral in 30 cases, circular in 3 cases). Distal femoral osteotomy was performed for distraction osteogenesis. The distraction rate was 1 mm/day divided into four 0.25 mm distractions. At external fixator removal, a prophylactic femoral nailing with Rush rod (preferred) or Titanium Elastic Nails (TEN) was performed. Patients’ characteristics are presented in Table 1.

### 2.2. Methods

The study is a retrospective cohort review. We have reviewed medical histories and radiograms. The patients were examined three times: (1) before hip reconstruction surgery, (2) before femoral lengthening, and (3) at follow-up. The patients underwent standard AP hip joints radiograms, internal rotation hip radiograms, Rippstein position radiograms [21], and standing lower limbs AP long cassette radiograms. We have measured the Center-Edge Angle (CEA, Wiberg), the Acetabular Inclination (AI), the Neck-Shaft Angle (NSA), and the Antetorsion Angle (AT), and the Shenton line was also evaluated [22,23,24,25,26,27]. Femoral torsion normal values were adapted from the Tönnis study [23], femoral retrotorsion was accepted as a negative true AT value, and diminished AT was accepted as <20 deg. Computed Tomography (CT) scans were available in all 6 patients with hip instability. The following parameters were measured on CT scans [18,28,29,30]: Acetabular Anteversion (AA), Axial Acetabular Index (AAI), and Acetabular Inclination in the anterior (AIa), middle (AIm), and posterior (AIp) part of the acetabulum. CT scans with plane presentation and measurements are shown in Figure 2 and Figure 3. Statistical analysis was performed with the use of STATISTICA v13.3 and PQStat v1.8.4. Significance was determined as *p* < 0.05.

## 3. Results

The patients’ recovery after SUPERhip preparatory surgery was uneventful. No infection occurred. In all 20 SUPERhip procedures, bone consolidation time was normal. In two cases of hip preparatory surgery limited to PFO and Dega osteotomy, an early hip plate destabilization occurred. In both cases, it was successfully treated with plate replacement. Both patients later developed hip joint instability.

### 3.1. Hip Stability during Femoral Lengthening

No case of hip instability occurred before or during femoral lengthening. All 33 patients (37 lengthenings) maintained clinically and radiologically stable hip joints at the examination before lengthening, during femoral lengthening (mean time of external fixator = 204 days, mean bone lengthening = 5.60 cm), and at the time of the external fixator removal.

### 3.2. Hip Stability after External Fixator Removal

Based on the criterion of 1 year follow-up after the external fixator removal, we analyzed 28 patients (30 lengthenings, 81%). We observed 6 patients (6/28, 21%) with hip joint instability. Hip instability was diagnosed at a mean of 637 days (1.74 years) after the external fixator removal, ranging from 127 to 1447 days.

We have noted four cases of hip subluxation and two cases of hip dislocation. In 5/6 patients, hip joint instability was diagnosed after the first lengthening; in one patient it was diagnosed after the second femoral lengthening. The patients surgically treated for hip instability finally presented with stable hip joints.

### 3.3. Hip Joint Morphology as Risk Factor for Hip Instability

#### 3.3.1. Initial Assessment

Based on the initial evaluation of the hip joint morphology we performed the analysis of 2 categories: (1) normal proximal femur morphology (univocal with Paley type 1a^1^), N = 13, and (2) abnormal proximal femur morphology (Coxa Vara and/or retrotorsion); this group comprised Paley types 1a^2^ and 1a^3^, N = 15. The incidence of hip instability after femoral lengthening was higher in the group with initially abnormal proximal femur morphology (33% vs. 8%), as shown in Table 2.

The hip joint morphological parameters measured at the initial assessment were analyzed with concern to the development of hip instability after femoral lengthening, Table 3. Diminished femoral antetorsion or femoral retrotorsion was widely observed among patients of our study (15/28, 54%) at the initial assessment. The incidence of abnormal femoral antetorsion deformity was higher in patients who later developed hip joint instability than in those who remained stable; however, it did not reach statistical significance (*p* = 0.099; OR 6). The incidence of Coxa Vara deformity was higher in patients who later developed hip joint instability than in those who remained stable; however, it was not statistically significant (*p* = 0.121; OR 4.28). The most common pathology in patients who later developed hip instability was abnormal femoral antetorsion (83%; 5/6 cases), then Coxa Vara (66%; 4/6 cases). Patients who developed hip joint instability presented a lower prevalence of AI and CEA abnormalities than those with stable hip joints.

#### 3.3.2. Before Lengthening Assessment

##### Shenton’s Line

In our cohort, 4/33 (14%) of patients presented the Shenton’s line disruption at the initial assessment, before hip preparatory surgery. After hip reconstruction surgery the patients presented radiologically stable hip joints without Shenton’s line disruption. None of these patients developed hip joint instability in later observation.

##### Center-Edge Angle, Acetabular Inclination, Neck-Shaft Angle and Antetorsion

The hip joint radiological parameters evaluated at pre- and post-hip preparatory surgery are shown in Table 4. Significant improvement in all parameters was noted *p* < 0.05.

Before lengthening, an evaluation showed no significant difference for CEA, NSA, and AT between patients who underwent the hip preparatory surgery and those who did not, *p* > 0.05. There was a statistically significant difference in AI, though both groups were within normal values (those with preparatory surgery had a lower mean AI than those without preparatory surgery).

At both the initial evaluation and the radiological hip joint evaluation before lengthening, no significant difference was noted between patients who later developed hip instability vs. those with stable hip joints, as shown in Table 5. A substantial difference (not statistically significant, *p* = 0.11) was noted for NSA, 124.9 deg vs. 134.1 deg.

The prevalence of hip joint abnormalities noted at the before lengthening assessment is shown in Table 6. The abnormalities observed at this stage were Coxa Vara and lower AT (<20 deg). Coxa Vara was shown to be indicative of hip instability development (50% in the instability group vs. 9% in the stable group; *p* < 0.05; OR 10, RR 5.5).

### 3.4. Parameters of Femoral Lengthening as Risk Factor for Hip Instability

The mean femoral lengthening was 5.7 cm, ranging from 1.8 cm to 7.8 cm. Mean femoral lengthening expressed as the femoral length percentage reached 22.8% (ranging from 4.5% to 40.2%). In 5/30 (16.7%) cases, the lengthening exceeded 30% of the femoral length; two of those patients developed hip joint instability later, and it was not statistically significant (*p* > 0.05). The femoral lengthening parameters concerning hip joint stability are presented in Table 7. The mean lengthening percentage in patients presenting late hip instability after femoral lengthening was 28.2% vs. 21.4% in patients who remained stable, the difference was not statistically significant (*p* = 0.08).

Before femur lengthening, the ratio of shorter femur length to healthy femur length was 0.80 for the whole group, and no significant difference between children with and children without late hip joint instability was found: 0.76 vs. 0.81, respectively, *p* > 0.05. After femur lengthening, the ratio revealed 0.94 for all children, and again, no difference was noted between children with versus without instability: 0.92 vs. 0.95, respectively, *p* > 0.05.

The mean Ex-Fix Index in our cohort was 41.1 days per 1 cm of lengthening, ranging from 24.8 to 135.8 days/cm (median 36.3 days/cm). No significant difference was found between patients with and patients without hip joint instability: 35.6 days/cm vs. 42.4 days/cm, *p* > 0.05.

### 3.5. Pelvic Osteotomy as a Risk Factor for Hip Instability

At the stage of the first surgery, consisting of hip joint reconstruction, and based on radiological criteria, 17/33 patients received pelvic osteotomy as a part of the procedure while the remaining 16/33 patients were not qualified for pelvic osteotomy. The number of patients who developed late hip joint instability was 4 out of 17 (24%) and 2 out of 16 (16.6%), respectively, a difference that was not statistically significant, *p* > 0.05.

### 3.6. CT Analysis of Hip Instability Cases

The CT scans were available only in patients with late hip joint instability and only at the stage where the subluxation/dislocation occurred. One patient could not be analyzed due to technical issues with the CT scan.

All five included patients presented a deficit of the coverage of the posterior part of the acetabulum. These patients presented a normal Acetabular Inclination angle on standard hip radiograms and a pathological high Acetabular Inclination angle measured in the posterior part of the acetabulum on CT scan (>25 deg); the difference between CT and RTG for the AI measured in the posterior part was statistically significant (28.3 deg vs. 19.8 deg, *p* < 0.01). Another finding was that the AI measured in the anterior, middle, and posterior parts of the acetabulum were significantly different in the CFD hip joint (16.2 deg vs. 22.4 deg vs. 28.3 deg, *p* < 0.01), with no such difference at the healthy side. The difference in AI measured in the posterior vs. anterior part of the acetabulum was 11.1 deg in the CFD unstable hip joint, and it was significantly higher than on the healthy side (11.1 deg vs. 3.0 deg, *p* < 0.05). CT analysis is shown in Table 8. A CT scan of a patient with hip instability is shown in Figure 2 and Figure 3.

### 3.7. Clinical Analysis of Hip Instability Cases

The key clinical information about hip instability cases is summarized in Table 9. Figure 4 shows radiograms and CT scans of the case of hip joint instability after femoral lengthening (Case 2).

## 4. Discussion

At the initial evaluation, all patients were classified as Paley type 1a of CFD, because there were no signs of femoral neck pseudoarthrosis or delayed consolidation of the proximal femur, and all patients presented a mobile hip and a mobile knee joint.

We have identified only a few studies investigating the topic of hip joint instability in Congenital Femoral Deficiency patients during femoral lengthening [8,9,10,11,12,13,14], and some of them have proposed risk factors (NSA < 120 deg, CEA < 20, AI > 25, CFD type 1b, proximal distraction osteotomy) that we have incorporated in treatment strategy at the department.

For all patients, a detailed clinical and radiological evaluation was performed, and completed with a comprehensive consultation. We have taken into consideration the known risks of hip instability in type 1a CFD patients. Therefore, we have undertaken preventive actions consisting of combining the SUPERhip procedure with valgus (in cases of Coxa Vara) and torsional femoral osteotomy with Dega iliac bone osteotomy (whenever qualified) [2,9]. Moreover, a distal femoral osteotomy was practiced as the site for distraction osteogenesis because such location results in a lower risk of hip joint instability during femoral lengthening [9]. Still, a considerable percentage (21%) of our patients developed hip joint instability. What is worth underlining, is the hip joint instability in the form of dislocation/subluxation, revealed late after completion of femoral lengthening. No dislocation/subluxation cases were present during femoral lengthening. In our cohort, hip joint instability developed long after the external fixator removal, usually more than one year later. The late hip joint instability risk has not been previously researched, so we feel that it is a strength of this study.

After hip preparatory surgery and before femoral lengthening, all patients presented a radiologically stable hip joint. However, even patients with normal-appearing radiological CEA and AI could develop late hip subluxation. Based on our cohort, we have shown that patients presenting a Coxa Vara, diminished femoral antetorsion, or femoral retrotorsion might be at higher risk of hip instability than those with normal AT and NSA: 33% vs. 8% of prevalence.

Musielak et al. postulated that acetabular dysplasia cannot be correctly evaluated on standard hip radiograms [18], and especially the posterior acetabulum deficit, which is considered a common deficit in CFD, cannot be evaluated. The authors postulate the superiority of CT scans in evaluating the deficit of the acetabulum in CFD patients. When generally approving such a statement, we need to mention the limitations of the CT scan, namely the radiation risk and the necessity of general anesthesia due to young patients. Therefore, we underline the need for the precise distinction of which Paley type 1a children will benefit the most from the CT hip evaluation. Based on our CFD type 1a cohort we would recommend the CT scans to be performed in every case of diminished femoral antetorsion or femoral retrotorsion, even with normal AI and CEA, and in every case of Coxa Vara.

Computed Tomography analysis has shown that type 1a_2_ and 1a_3_ (diminished antetorsion or retrotorsion) CFD hip joints presented a posterolateral deficit of the acetabulum, which is consistent with the literature [17,18,31]. Such acetabular morphology needed adjusted surgical technique to improve posterior acetabulum coverage. The modification of classic Dega iliac osteotomy consists of reorientation of the bony cut to achieve better mobilization of the posterior acetabular fragment. We have successfully used this technique in the treatment of five patients with hip instability and adopted it for preparatory surgeries in CFD. On the other hand, one hip joint dislocation occurred in type 1a_1_ (normal AT, NSA, AI, and CEA); the CT revealed a lateral deficit of the acetabulum. We suppose the hip joint dislocation in Paley type 1a_1_ probably results from a different mechanism than in type 1a_2_ or 1a_3_ and can be treated with classic Dega osteotomy.

The amount of femoral lengthening has been proposed as a factor of hip joint instability [32]. In our study, femoral lengthening parameters between unstable and stable groups did not differ significantly. Worth noting is the higher mean lengthening (28.2% vs. 21.4%, *p* = 0.08) in patients who later developed hip joint instability. However, only two patients in the instability group exceeded 30% of mean lengthening (Table 6) and the rest of them had a lengthening percentage consistent with the stable group. Due to the small group size, we could not draw conclusions considering the amount of femoral lengthening.

In CFD children, the affected side shortening can be attributed to either femoral bone shortening or proximal femur deformity [2,33]. In our cohort, the femoral shortening ratio has not shown a correlation with hip joint instability risk during or after femoral lengthening. Contrarily, the degree of proximal femoral deformity has been shown to affect the risk of hip joint instability after femoral lengthening. Previous studies reported that Paley type 1a CFD remains relatively safe from hip dislocation during femur lengthening [9]. However, we have demonstrated that late hip instability may occur, not recognizable during or shortly after the end of femoral lengthening. According to our analysis, the torsional proximal femur deformity (diminished antetorsion or retrotorsion) and lower Neck-Shaft Angle (Coxa Vara) might be risk factors for late hip instability. Therefore, utilizing subclasses of Paley 1a classification and evaluating femur torsion seems crucial in managing these patients.

There are some strengths and limitations to this study. Due to the rarity of CFD, all the published studies faced two similar limitations: (1) non-homogenous surgical technique and (2) small group size. In this study, we managed to escape the first and to diminish the second limitation. (1) All patients of our cohort were treated with the same surgical technique (SUPERhip procedure comprising PFO and Dega osteotomy, whenever qualified), performed at the same department, and by the same surgeon (M.S.) experienced in limb reconstruction and lengthening. (2) A small group size remains a limitation of this study; however, it needs to be taken into consideration that all the children in our cohort presented the same CFD type—Paley type 1a. Also, the cohort comprised all of the consecutive CFD children who qualified for reconstructive lower limb treatment and were admitted to our department. Another limitation of this study was the availability of the post-lengthening CT scans in patients without hip joint instability. We have not routinely performed CT due to high radiation and the necessity of general anesthesia. In future studies, hip joint assessment based on a 3D MRI scan could overcome the limitations of this study.

We believe that reconstructive hip joint surgery in CFD can delay the early hip joint degeneration up to the moment that Total Hip Arthroplasty is necessary to improve the quality of life by reducing pain and improving function [34].

## 5. Conclusions

Femur lengthening can be successfully performed in Paley type 1 CFD children while maintaining hip joint stability. Late hip joint instability may occur after femoral lengthening despite hip morphology appearing to be normal on radiograms at initial evaluation and after femoral lengthening. Subluxation or dislocation may occur, even up to 2 years after external fixator removal. Undercorrected Neck-Shaft Angle was a risk factor of hip joint instability in CFD patients subjected to femoral lengthening. Femoral retrotorsion or diminished antetorsion might be indicative of acetabular dysplasia with the posterior deficit, even with quasi-normal proximal femur morphology in standard AP hip projection. Patients with abnormal AT might be at higher risk of hip joint instability after femoral lengthening.

## Figures and Tables

**Figure 1 children-11-00500-f001:**
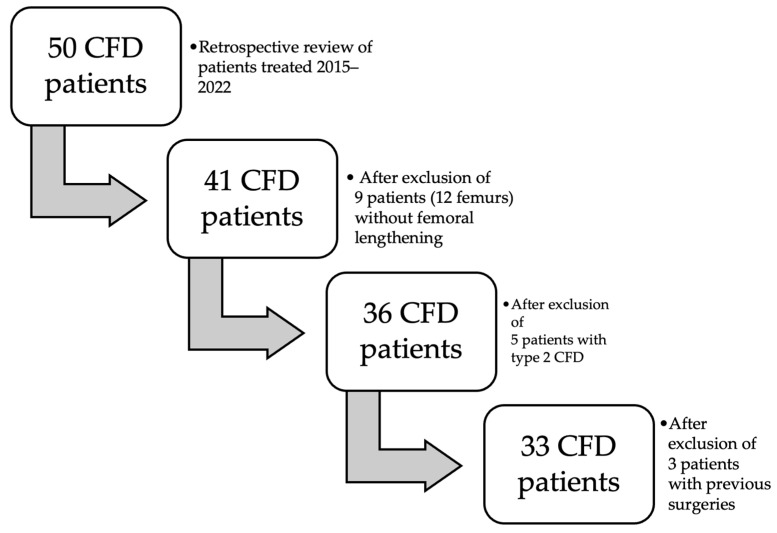
Study group recruitment.

**Figure 2 children-11-00500-f002:**
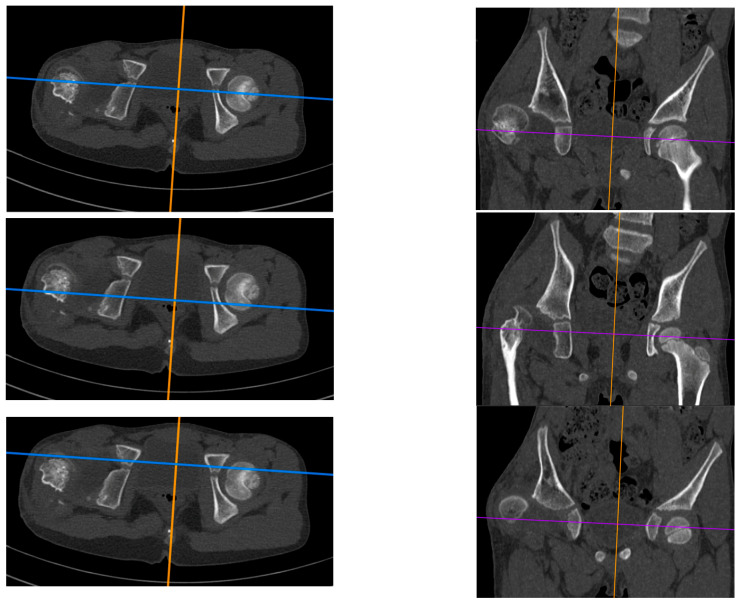
CT examination of patient with right hip dislocation. Left: transverse view, right: coronal view. Evaluation of medial (**first row**), posterior (**middle row**), and anterior (**last row**) parts of the acetabulum.

**Figure 3 children-11-00500-f003:**
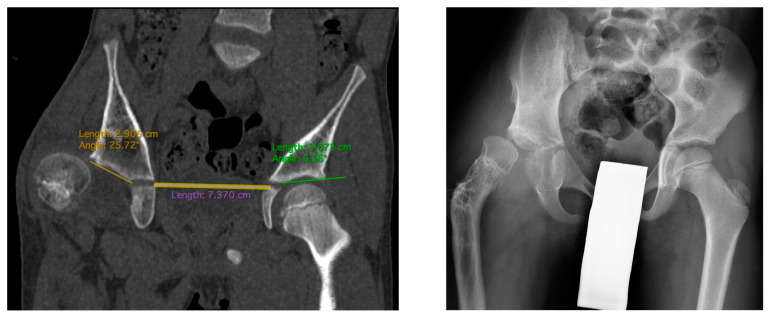
CT scan showing AIm measurements (**left**): 24 deg on the affected side vs. 6 deg on the healthy side. Hip radiogram of the same patient with hip joint dislocation (**right**).

**Figure 4 children-11-00500-f004:**
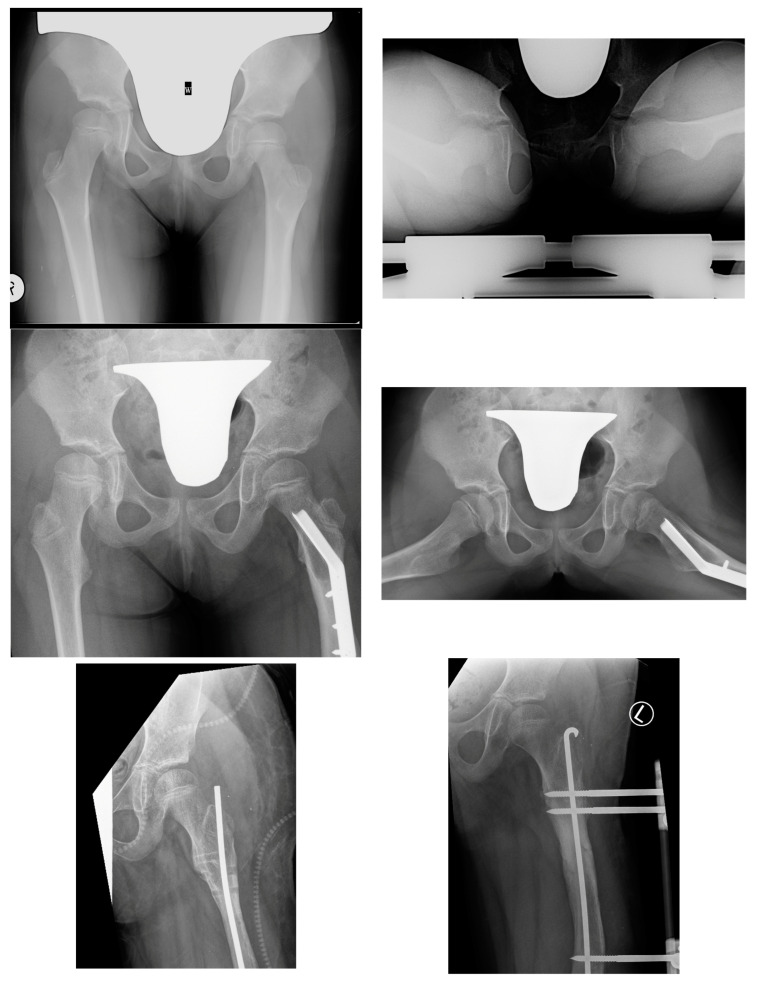
Patient with hip joint instability after femoral lengthening. The first row shows radiograms at initial evaluation with diminished antetorsion on the right radiogram. The second row shows radiograms after hip preparatory surgery. The third row shows radiograms after removal of the external fixator (left side) and after treatment of the femoral fracture—Rush rod with external fixator (right side). The fourth row shows radiograms the first post-op day after removal of the Rush rod, with visible left hip joint dislocation. The final and fifth row shows radiograms 1 year after surgical treatment—the left hip joint remained stable (last follow-up—over 4 years post-surgical treatment of dislocation).

**Table 1 children-11-00500-t001:** Patients’ characteristics.

	SUPERhip Prior to Lengthening	No SUPERhip Prior to Lengthening	Total (Percentage)
Group characteristics	N = 20	N = 13	N = 33 (100)
Gender			
male	8	8	16 (48%)
female	12	5	17 (52%)
Side of deformity			
left	11	7	18 (55%)
right	9	6	15 (45%)
Congenital Femoral Deficiency Paley type [2]			
1a_1_	5	11	16 (48%)
1a_2_	2	0	2 (6%)
1a_3_	13	2	15 (45%)
Age at the time of hip reconstruction (years, range)	4.3 (2.7–8.1)		
Age at the time of femoral lengthening (years, range)	6.6 (4.3–12.3)	9.6 (6.9–14.3)	7.8 (4.3–14.3)
Additional diagnosis of fibular hemimelia Achterman and Kalamchi type [21]	14	10	24 (73%)
1a	8	6	14 (42%)
1b	2	2	4 (12%)
2	4	2	6 (18%)
Follow-up in months (mean, range)			
After SUPERhip	70 (24–100)		
After end of femoral lengthening	39 (11–73)	39 (12–106)	39 (11–106)

**Table 2 children-11-00500-t002:** Hip joint instability in relation to proximal femur morphology at initial assessment.

	Normal Proximal Femur Morphology N = 13	Abnormal Proximal Femur Morphology N = 15	Total N = 28
Hip instability number (%)	1 (8%)	5 (33%)	6 (21%)
Coxa Vara (NSA < 120 deg)	0 (0%)	11 (73%)	11 (39%)
Retrotorsion (<0 deg)	0 (0%)	7 (44%)	7 (25%)
Diminished antetorsion (<20 deg)	0 (0%)	8 (53%)	8 (29%)
Abnormal antetorsion (retro- or diminished)	0 (0%)	15 (100%)	15 (54%)
AI > 25 deg.	0 (0%)	5 (33%)	5 (18%)
CEA < 20 deg	6 (46%)	8 (53%)	14 (50%)

**Table 3 children-11-00500-t003:** Hip joint morphological parameters (measured at the initial assessment) in relation to development of hip instability after femoral lengthening.

	Instable Hip Joint N = 6	Stable Hip Joint N = 22	Total N = 28	*p*	OR
Coxa Vara (NSA < 120 deg)	4 (66%)	7 (32%)	11	*p* = 0.121	4.28
Retrotorsion (<0 deg)	2 (33%)	5 (23%)	7	*p* = 0.595	1.7
Diminished antetorsion (<20 deg)	3 (50%)	5 (23%)	8	*p* = 0.190	3.4
Abnormal antetorsion (<20 deg or <0 deg)	5 (83%)	10 (45%)	15	*p* = 0.099	6
AI > 25 deg	1 (17%)	4 (18%)	5	*p* = 0.932	0.9
CEA < 20 deg	2 (33%)	12 (55%)	14	*p* = 0.357	0.42

**Table 4 children-11-00500-t004:** Hip joint radiological parameters pre- and post- hip preparatory surgery.

	With Preparatory Surgery			Without Preparatory Surgery	
	Initial Assessment (N = 22) Mean (SD)	Before Lengthening (N = 22) Mean (SD)	Initial vs. before Lengthening *p*	Before Lengthening (N = 11) Mean (SD)	Before Lengthening with vs. without Preparatory Surgery, *p*
Neck-Shaft Angle	117.8 (26.7)	133.5 (13.2)	*p* < 0.01 *	136.2 (6.0)	*p* = 0.43
Antetorsion	7.0 (15.7)	31.8 (16.0)	*p* < 0.01 *	28.5 (10.6)	*p* = 0.74
Acetabular Inclination	20.0 (7.7)	11.2 (4.3)	*p* < 0.01 *	14.7 (5.0)	*p* < 0.05 *
Center-Edge Angle	18.9 (7.4)	24.7 (4.5)	*p* < 0.01 *	22.3 (6.3)	*p* = 0.17

* statistically significant.

**Table 5 children-11-00500-t005:** Initial assessment vs. before lengthening hip joint parameters in relation to development hip joint instability.

	Hip Joint Instability (N = 6)		Stable Hip Joint (N = 22)	
	Initial	Before Lengthening	Initial	Before Lengthening
Neck-Shaft Angle	112.7 (10.8)	124.9 (14.2)	123.2 (4.9)	134.1 (11.8), *p* = 0.11
Antetorsion	6.55 (22.0)	31.6 (20.2)	14.2 (19.8)	29.9 (13.4)
Acetabular Inclination	16.5 (7.1)	11.7 (5.5)	17.7 (6.6)	12.8 (5.4)
Center-Edge Angle	23.4 (6.5)	26.1 (4.4)	20.4 (7.6)	23.7 (5.5)

**Table 6 children-11-00500-t006:** Hip joint morphological parameters (measured at before lengthening assessment) in relation to development of hip instability after femoral lengthening.

	Instable Hip Joint N = 6	Stable Hip Joint N = 22	Total N = 28	*p*	OR
Coxa Vara (NSA < 120 deg)	3 (50%)	2 (9%)	5	*p* < 0.05 *	10
Retrotorsion (<0 deg)	0 (0%)	0 (0%)	0		
Diminished antetorsion (<20 deg)	2 (33%)	5 (23%)	7	*p* = 0.59	
Abnormal antetorsion (<20 deg or <0 deg)	2 (33%)	5 (23%)	7	*p* = 0.59	
AI > 25 deg	0 (0%)	0 (0%)	0		
CEA < 20 deg	0 (0%)	7 (32%)	7		

* statistically significant.

**Table 7 children-11-00500-t007:** Femoral lengthening parameters in relation to hip joint stability.

	Instable Hip Joint N = 6 (6 Lengthenings)	Stable Hip Joint N = 22 (24 Lengthenings)	*p*	Total N = 28 (30 Lengthenings)
Lengthening (cm)	6.4 (1.0)	5.5 (1.4)	*p* = 0.17	5.7 (1.4)
Lengthening (%)	28.2 (0.1)	21.4 (0.1)	*p* = 0.08	22.8 (0.1)
Shorter to healthy femur ratio before lengthening	0.76 (0.08)	0.81 (0.08)	*p* = 0.14	0.80 (0.08)
Shorter to healthy femur ratio after lengthening	0.92 (0.07)	0.95 (0.08)	*p* = 0.43	0.94 (0.08)
Ex-Fix Index (days/cm)	35.6 (5.3)	42.4 (23.1)	*p* = 0.85	41.1 (20.9)
Ex-Fix duration (in days)	227 (37)	209 (35)	*p* = 0.30	212 (35)

Values presented as mean (SD).

**Table 8 children-11-00500-t008:** CT analysis of five cases of posterior hip joint instability in CFD.

	CFD Side	Healthy Femur	*p*
AIa	16.2 (9.0)	6.5 (2.7)	*p* < 0.01 *
AIm	22.4 (3.9)	6.7 (3.1)	*p* < 0.01 *
AIp	28.3 (5.6)	9.9 (2.1)	*p* < 0.01 *
Acetabular Anteversion	10.5 (11.5)	11.1 (6.0)	*p* = 0.81
Axial Acetabular Inclination	106.7 (26.0)	109.1 (13.0)	*p* = 0.83
Difference between AI in anterior and posterior part of acetabulum	11.1 (6.1)	3.0 (3.4)	*p* < 0.01 *

* statistically significant.

**Table 9 children-11-00500-t009:** The clinical information about hip instability cases.

Case	Hip Joint Initial Morphology	Hip Preparatory Surgery	Complications after Hip Preparatory Surgery	External Fixator Type for Lengthening	Mean Lengthening (%)	Lengthening Complications	When Instability Was Diagnosed (Days after External Fixator Removal)
1	Normal	SUPERhip	no	Monolateral	23.2%	no	184 days
2	Diminished antetorsion	SUPERhip	no	Monolateral	21.2%	Femoral fracture after frame removal	801 days
3	Coxa Vara, Diminished antetorsion	SUPERhip + Dega	no	Monolateral	39.7%	no	127 days
4	Coxa Vara, Retrotorsion	SUPERhip + Dega	no	Monolateral	40.2%	no	509 days
5	Coxa Vara, Retrotorsion	PFO + Dega	Plate aseptic loosening, treated with plate replacement and fixation.	Monolateral	16.8%	no	752 days
6	Coxa Vara, Diminished antetorsion	PFO + Dega	Plate aseptic loosening, treated with plate replacement and fixation.	Circular	27.8%	Femoral fracture after prophylactic nail removal	1447 days

## Data Availability

The data presented in this study are available on request from the corresponding author due to ethical and privacy reason.

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
