# Peer review of "Hip Joint Stability during and after Femoral Lengthening in Congenital Femoral Deficiency"

_children, 2024, doi:10.3390/children11040500_

Round 1

Reviewer 1 Report

Comments and Suggestions for Authors

This manuscript is retospective study aimed review hip stability in Paley type 1a CFD patients undergoing femoral lengthening. 33 patients with unilateral CFD were treated between 2014 and 2023. In 20/33 cases (60.6%) the SUPERhip preparatory surgery was performed at mean age of 4.3 yo (range 2.7- 8.1). The femoral lengthening using an external fixator was performed at 7.8 yo (range 4.3-14.3).  All patients presented a stable hip joint after preparatory surgery and during femoral lengthening. Six cases of hip instability at mean 637 days after the external fixator removal were  observed (range 127 to 1447 days). No significant differences between stable vs. unstable hips were noted for: (1) Center-Edge Angle: 23.7 vs 26.1 deg; (2) Acetabular Inclination 12.8 vs 11.7 deg; and (3) Ex-Fix Index: 35.6 days/cm vs. 42.4 days/cm; p>0.05. Late hip instability was related to Coxa Vara and decreased femoral antetorsion before lengthening.

I read the article with interest, the title is well thought out and faithfully reflects the content of the study.

A)   The abstract is sufficiently developed, and it is useful to frame the purpose of the study, but a few concerns are present:

Comment 1: A brief mention of the characteristics of the study should be write.

B)    In the introduction, the characteristics of the Congenital Femoral Deficiencyhave been sufficiently described.

Comment 2: “The clinical presentation is heterogeneous it may be isolated defect or accompany other deformities; unilateral or bilateral; with varying severity from mild shortening affecting the proximal part of the femur to severe, including absence of the femur bone. In its unilateral form, due to limb length discrepancy CFD is treated with limb lengthening.” Adding some bibliographic references about it.

C)    The materials and methods have been shortly developed.

Comment 3: Who did the diagnose? Was he a pediatric orthopedist?

D)   The discussion is sufficiently developed.

Comment 7: You did not specify the limitations of your study and what could be done in subsequent studies on the topic to overcome the limitations of this study.

Comment 8: It would be appropriate to specify what could be done to cure the outcomes of DDH, adding an appropriate bibliographical reference, for example: (De Salvo S. et al (2024) " Total hip arthroplasty in patients with common pediatric hip orthopedic pathology

Comment 9: In the conclusions you insert recommendations that you did not implement in your study, it would be appropriate to define and describe them carefully.

Finally, English language editing is needed.

Nevertheless, some minor changes are needed to be considered suitable for publication.

Comments on the Quality of English Language

 English language editing is needed

Author Response

Thank you for giving us the opportunity to submit a revised draft of the manuscript “Hip joint stability during and after femoral lengthening in Congenital Femoral Deficiency” for publication in the “Children”. We appreciate the time and effort that you have dedicated to providing feedback on our manuscript and are grateful for the insightful comments on and valuable improvements to our paper. We have incorporated the suggestions made by you. The changes are highlighted within the revised manuscript. Please see below, for a point-by-point response to your comments and concerns. All page numbers refer to the revised manuscript file.

Reviewer comment to the Author:

This manuscript is retospective study aimed review hip stability in Paley type 1a CFD patients undergoing femoral lengthening. 33 patients with unilateral CFD were treated between 2014 and 2023. In 20/33 cases (60.6%) the SUPERhip preparatory surgery was performed at mean age of 4.3 yo (range 2.7- 8.1). The femoral lengthening using an external fixator was performed at 7.8 yo (range 4.3-14.3).  All patients presented a stable hip joint after preparatory surgery and during femoral lengthening. Six cases of hip instability at mean 637 days after the external fixator removal were  observed (range 127 to 1447 days). No significant differences between stable vs. unstable hips were noted for: (1) Center-Edge Angle: 23.7 vs 26.1 deg; (2) Acetabular Inclination 12.8 vs 11.7 deg; and (3) Ex-Fix Index: 35.6 days/cm vs. 42.4 days/cm; p>0.05. Late hip instability was related to Coxa Vara and decreased femoral antetorsion before lengthening.

I read the article with interest, the title is well thought out and faithfully reflects the content of the study.

  1. A) The abstract is sufficiently developed, and it is useful to frame the purpose of the study, but a few concerns are present:

Comment 1: A brief mention of the characteristics of the study should be write.

Author response: Thank you for pointing this out. We have included information about the characteristics of the study in the abstract. See the revised text on page 1.

  1. B) In the introduction, the characteristics of the Congenital Femoral Deficiency have been sufficiently described.

Comment 2: “The clinical presentation is heterogeneous it may be isolated defect or accompany other deformities; unilateral or bilateral; with varying severity from mild shortening affecting the proximal part of the femur to severe, including absence of the femur bone. In its unilateral form, due to limb length discrepancy CFD is treated with limb lengthening.” Adding some bibliographic references about it.

Author response: Thank you for this comment. We have added bibliographic references about the characteristics of the CFD. See the revised text on page 1.

  1. C) The materials and methods have been shortly developed.

Comment 3: Who did the diagnose? Was he a pediatric orthopedist?

Author response: As suggested, we have included information about the diagnosis of patients included in our study group. See the revised text on page 2.

  1. D) The discussion is sufficiently developed.

Comment 7: You did not specify the limitations of your study and what could be done in subsequent studies on the topic to overcome the limitations of this study.

Author response: Thank you for this suggestion, we have improved the discussion section of the article and specified the limitations of the study. We have included our opinion on overcoming them in future studies. See revised text on page 14.

Comment 8: It would be appropriate to specify what could be done to cure the outcomes of DDH, adding an appropriate bibliographical reference, for example: (De Salvo S. et al (2024) " Total hip arthroplasty in patients with common pediatric hip orthopedic pathology.

Author response: Thank you for this suggestion for the discussion. We have added our opinion on early hip joint degeneration in CFD and the need for THA in those patients. See the revised text on page 15.

Comment 9: In the conclusions you insert recommendations that you did not implement in your study, it would be appropriate to define and describe them carefully.

Author response: Thank you for this concern. We have revised our conclusion section to fit appropriately with our study results. See the revised text on page 15. We have discussed the recommendation of CT scans in CFD patients in the discussion section of the manuscript. See the revised text on page 14.

Finally, English language editing is needed.

Author response: We have performed language editing of the manuscript.

Nevertheless, some minor changes are needed to be considered suitable for publication.

Author response: Thank you for your comments. We believe those have benefited the current manuscript.

Reviewer 2 Report

Comments and Suggestions for Authors

The paper presented for the reviewing is aimed to review a cohort of homogenous Paley type 1a CFD patients in order to analyze the hip joint stability during and after femoral lengthening.
The authors analyzed their relatively large and consistent group of patients with congenital, femoral deficiency treated by femoral lengthening Retrospective analysis of the cohort demonstrated substantial proportion of hip instability after the lengthening procedure (6 of 28 patients). The research question postulated by the authors was to analyze the possible causes of hip joint instability and its management.
To answer the research question, the authors conducted comparative analysis of the radiographic parameters of the hips. Among those patients operated by femoral lengthening procedure and compatible with the inclusion criteria, 6 patients developed hip joint instability in the midterm follow-up period after the completion of the lengthening. The comparative analysis of the hip joint morphology as the risk factor for instability was done at the initial preoperative point, after the lengthening, and 1 year after the device removal. The findings demonstrated the differences between stable and unstable groups. The second important part of the study was to, assess their differences in radiographic and tomographic acetabular indices as the predictive factors for secondary hip instability. Tomographic measurements of acetabular index in three points within the coronal plane were done only in the 5 patients of the unstable group. So, comparative analysis was done between unstable and the contralateral hip. There was no comparative data available with the stable group. The most important information is well presented in the tables.
Despite the obvious advantages of the study, including relatively large number of patients with rare condition, homogeneous group, well-documented follow up etc., there are many obstacles, which should be modified, explained or withdrawn.
The formal aim of the study was to analyze the possible causes of hip joint instability and its management. In fact, the authors just offered the data and the interpretation of these findings without any chance to prove the predictive effect of those factors. Presentation of the result is more or less consistent with the research questions excepting some minor inaccuracies like CEA, instead of NSA in the line 217, page 7.
The most doubts are in regard to conclusions, following the discussion chapter.
Conclusion 1. I fact, 2 of 6 patients (33%) developed instability with the lengthening less than 22%.
Conclusion 2. It is obvious that in congenital femoral deficiency proximal femur morphology is abnormal by definition, it is just possible to say about normal or quasi-normal formal radiographic hip parameters.
Conclusion 3. Coxa Vara deformity was not indicative for instability risk. The authors noticed that the incidence of Coxa Vara deformity was higher, but not statistically significant.
Conclusion 4. In fact, only the statement that abnormal torsion might be risk factor of hip joint instability after the femoral lengthening is based on the results of the study. The other sentences are just the part of the discussion.
Conclusion 5 is also more related to the discussion because there was no comparative analysis between stable and unstable group.
It is also important to notice that all the patients who developed instability underwent proximal femoral osteotomies, and most - acetabular correction. That means that Coxa Vara and retrotorsion actually were undercorrected at the previous surgeries that should be definitely stressed out in the discussion and/or recommendations.
The figures are more related to the methodology of measurements, then to illustration of the clinical practice. Taking into account the small number of patients (six unstable hips), maybe it would be more logical to accompany the text with more demonstrative figures (at least of most demonstrative cases before treatment, after treatment, and after diagnosis of instability) but it is on the authors’ choice. The only case of hip instability presented in the paper demonstrates normal NSA in contrast to the previously discussed points.
The paper is based on the unique and very interesting material, but needs substantial modifications before publication.

Comments on the Quality of English Language

The English text contains many minor inaccuracies that require editing.

Author Response

Thank you for giving us the opportunity to submit a revised draft of the manuscript “Hip joint stability during and after femoral lengthening in Congenital Femoral Deficiency” for publication in the “Children”. We appreciate the time and effort that you have dedicated to providing feedback on our manuscript and are grateful for the insightful comments on and valuable improvements to our paper. We have incorporated the suggestions made by you. The changes are highlighted within the revised manuscript. Please see below, for a point-by-point response to your comments and concerns. All page numbers refer to the revised manuscript file.

Reviewer comment to the Author:

The paper presented for the reviewing is aimed to review a cohort of homogenous Paley type 1a CFD patients in order to analyze the hip joint stability during and after femoral lengthening.

The authors analyzed their relatively large and consistent group of patients with congenital, femoral deficiency treated by femoral lengthening Retrospective analysis of the cohort demonstrated substantial proportion of hip instability after the lengthening procedure (6 of 28 patients). The research question postulated by the authors was to analyze the possible causes of hip joint instability and its management.

To answer the research question, the authors conducted comparative analysis of the radiographic parameters of the hips. Among those patients operated by femoral lengthening procedure and compatible with the inclusion criteria, 6 patients developed hip joint instability in the midterm follow-up period after the completion of the lengthening. The comparative analysis of the hip joint morphology as the risk factor for instability was done at the initial preoperative point, after the lengthening, and 1 year after the device removal. The findings demonstrated the differences between stable and unstable groups. The second important part of the study was to, assess their differences in radiographic and tomographic acetabular indices as the predictive factors for secondary hip instability. Tomographic measurements of acetabular index in three points within the coronal plane were done only in the 5 patients of the unstable group. So, comparative analysis was done between unstable and the contralateral hip. There was no comparative data available with the stable group. The most important information is well presented in the tables.

Despite the obvious advantages of the study, including relatively large number of patients with rare condition, homogeneous group, well-documented follow up etc., there are many obstacles, which should be modified, explained or withdrawn.

The formal aim of the study was to analyze the possible causes of hip joint instability and its management. In fact, the authors just offered the data and the interpretation of these findings without any chance to prove the predictive effect of those factors. Presentation of the result is more or less consistent with the research questions excepting some minor inaccuracies like CEA, instead of NSA in the line 217, page 7.

Author response: Thank you for pointing this out. We have corrected these inaccuracies. See the revised text on page 8.

The most doubts are in regard to conclusions, following the discussion chapter.

Conclusion 1. I fact, 2 of 6 patients (33%) developed instability with the lengthening less than 22%.

Author response: We agree with the reviewer’s assessment. We have removed this conclusion and added a section in the discussion chapter of the manuscript that concerns the amount of femoral lengthening as a risk factor for hip joint instability. We have added a bibliographic reference. See the revised text on pages 14 and 15.

Conclusion 2. It is obvious that in congenital femoral deficiency proximal femur morphology is abnormal by definition, it is just possible to say about normal or quasi-normal formal radiographic hip parameters.

Author response: Thank you for this suggestion. We have revised the second conclusion. The formal radiographic quasi-normal appearance of the proximal femoral morphology in CFD does not represent true morphology. Thus, we have discussed recommendations for performing CT scans in CFD in the discussion chapter. See the revised text on pages 14 and 15.

Conclusion 3. Coxa Vara deformity was not indicative for instability risk. The authors noticed that the incidence of Coxa Vara deformity was higher, but not statistically significant.

Author response: We appreciate the reviewer’s feedback. As shown in Table 5, the difference in NSA before lengthening was not significant, but substantial (124.9 deg vs 134.1 deg) comparing unstable and stable groups. However, the incidence of NSA < 120 deg. (Coxa Vara) deformity before lengthening evaluation was higher in the unstable group, and it was statistically significant (p<0.05). We have revised the third conclusion to highlight the risk of undercorrection of NSA at the preparatory surgery. See the revised text on page 15.

Conclusion 4. In fact, only the statement that abnormal torsion might be risk factor of hip joint instability after the femoral lengthening is based on the results of the study. The other sentences are just the part of the discussion.

Conclusion 5 is also more related to the discussion because there was no comparative analysis between stable and unstable group.

Author response: As suggested by the reviewer, we have revised the fourth conclusion. We have also withdrawn the fifth conclusion. There is a section where we discuss the recommendation for CT in the discussion chapter. See the revised text on pages 14 and 15.

It is also important to notice that all the patients who developed instability underwent proximal femoral osteotomies, and most - acetabular correction. That means that Coxa Vara and retrotorsion actually were undercorrected at the previous surgeries that should be definitely stressed out in the discussion and/or recommendations. 

Author response: We agree with the reviewer’s assessment. As suggested by the reviewer, we have revised and addressed the undercorrection in the conclusion chapter. See the revised text on page 15.

The figures are more related to the methodology of measurements, then to illustration of the clinical practice. Taking into account the small number of patients (six unstable hips), maybe it would be more logical to accompany the text with more demonstrative figures (at least of most demonstrative cases before treatment, after treatment, and after diagnosis of instability) but it is on the authors’ choice. The only case of hip instability presented in the paper demonstrates normal NSA in contrast to the previously discussed points. 

Author response: We think this is an excellent suggestion. We have moved figures 2 and 3 into the methodology chapter, where they belong. As suggested, we have added some more demonstrative figures that represent patients who developed hip joint instability. See revised figures on pages 5, 6, 12, and 13.

The paper is based on the unique and very interesting material, but needs substantial modifications before publication.

Author response: Thank you for your comments. We believe those have benefited the current manuscript.

Comments on the Quality of English Language

The English text contains many minor inaccuracies that require editing.

Author response: We have performed language editing of the manuscript.

Round 2

Reviewer 2 Report

Comments and Suggestions for Authors

The authors reworked the paper, according to the previous recommendation. The paper is recommended for the publication in the present version.